# In silico evaluation of garlic-derived organosulfur compounds as multi-target inhibitors of breast cancer biomarkers

Courage Siame[ID][1], Benedict Ofori[ID][2], Lily Paemka[3,4], Kwabena Owusu Danquah[ID][5*]

1 Department of Immunology, Noguchi Memorial Institute for Medical Research, College of Health Sciences, University of Ghana, Legon, Accra, Ghana, 2 Medical College of Georgia, The Graduate School, Augusta University, Augusta, Georgia, United States of America, 3 West African Centre for Cell Biology of Infectious Pathogens (WACCBIP), Department of Biochemistry, Cell and Molecular Biology, University of Ghana, Accra, Ghana, 4 West African Genetic Medicine Centre, University of Ghana, Accra, Ghana, 5 Department of Clinical Pathology, Noguchi Memorial Institute for Medical Research, College of Health Sciences, University of Ghana, Legon, Accra, Ghana

* kdanquah@noguchi.ug.edu.gh

## Abstract

Breast cancer is the leading cause of cancer mortality among women globally, and drug resistance complicates treatment. Garlic-derived organosulfur compounds exhibit anticancer potential, but their multi-target activity against key breast cancer biomarkers remains unclear. This study utilized AutoDock Vina for molecular docking, OpenBabel for post-docking energy minimization, and employs SWISS-ADME and PreADMET platforms for ADMET profiling to assess six garlic compounds (Z-ajoene, allyl-methyl trisulfide, diallyl disulfide, diallyl sulfide, diallyl trisulfide, and S-allyl-L-cysteine) against clinically relevant breast cancer targets. Z-ajoene showed strong binding to Bcl-2, Topoisomerase II, and CDK-2, while S-allyl-L-cysteine targets five biomarkers. All compounds complied with Lipinski's rule of five, indicating good oral bioavailability, and display favorable ADMET properties with no mutagenic or tumorigenic risks. Most compounds were predicted to inhibit P-glycoprotein, while only Z-ajoene showed potential inhibition of CYP2C9, suggesting possible drug-drug interactions. Despite moderate affinities, these compounds may serve as potential promising multi-target agents in breast cancer therapy. Our computational findings provide preliminary evidence that garlic-derived compounds warrant further in vitro and in vivo evaluation, particularly in the context of drug-resistant breast cancer.

## Introduction

Breast cancer is the most frequently diagnosed and leading cause of cancer-related deaths among women worldwide [1]. The most recent available data indicate a growing global burden of breast cancer, with an estimated 2.30 million incident cases, 764,000 deaths, and 24.1 million disability-adjusted life years among females

**Data availability statement:** All relevant data are within the manuscript and its Supporting Information files.

**Funding:** The author(s) received no specific funding for this work.

**Competing interests:** The authors have declared that no competing interests exist.

[2]. Current projections estimate that by 2050, breast cancer will account for approximately 3.2 million new cases and 1.1 million deaths annually worldwide, with the greatest increases expected in countries with a low Human Development Index [3]. The development of breast cancer arises from a complex interplay of environmental and genetic factors that leads to ductal or lobular hyperproliferation, eventually conferring metastatic potential and contributing to its high lethality [4,5]. Current treatment strategies primarily involve targeted chemotherapy that utilizes agents directed against specific biomarkers associated with key cancer hallmarks. However, the rise in multidrug resistance has significantly reduced treatment effectiveness, causing therapeutic failure in many patients [6,7]. Multidrug resistance in breast cancer arises through various mechanisms, including enhanced drug metabolism, increased efflux pump activity, altered growth factor signaling, upregulated DNA repair, genetic mutations, and epigenetic modifications [6]. The progression and survival of breast cancer cells are further driven by dysregulated molecular pathways, including evasion of apoptosis through proteins like B-cell lymphoma 2 (Bcl-2) and X-linked inhibitor of apoptosis protein (XIAP), unchecked proliferation via cyclin-dependent kinases (CDKs) and topoisomerases, and tumor-induced angiogenesis mediated by vascular endothelial growth factor and its receptor (VEGF/VEGFR2) signaling [8–15]. These macromolecules are currently targeted by existing breast cancer therapies hence identification of more efficacious agents against them could be very beneficial in treating resistant breast cancer types. An emerging therapeutic target is the guanine quadruplex (G4) DNA structure, which is abundant in the promoter regions of oncogenes such as cellular myelocytomatosis oncogene (*c-MYC*), where stabilization could inhibit oncogenic transcription [16–19]. Hence, there is a pressing need for novel therapeutic agents with greater specificity, lower toxicity, and improved resistance profiles. Current efforts include screening plant-derived compounds to combat the rising drug resistance, as over 64.9% of drugs in clinical use for cancer are derived from plant sources [20,21].

*Allium sativum L*. (garlic) belongs to the family Amaryllidaceae, contains 65% water, and 2.3% organosulfur compounds [22]. It is one of the many plants grown for food and used traditionally as medicine. Garlic-derived compounds have demonstrated anticarcinogenic, antimutagenic, bacteriostatic, hypoglycemic, and anticancer activities in in vitro and in vivo models [22–26]. In breast cancer, several garlic-derived compounds have demonstrated cytotoxic and antiproliferative effects. Diallyl trisulfide has shown cytotoxicity against MCF-7 and MDA-MB-468 cell lines [27], while S-allyl-L-cysteine and ajoene reduced proliferation and induced growth arrest in MCF-7, and MDA-MB-231 cells, respectively [26]. Diallyl disulfide is also known to inhibit the growth of multiple breast cancer cell lines, including MDA-MB-231, KPL-1, MKL-F, and MCF-7, and suppress mammary tumor development in rats [26,27–28]. These anti-breast cancer effects highlight the strong potential of sulfur-containing garlic-derived compounds for future breast cancer therapy. However, compounds such as Z-ajoene, allyl-methyl trisulfide, diallyl disulfide, diallyl sulfide, diallyl trisulfide, and S-allyl-L-cysteine remain underexplored regarding their mechanisms of action, anti-breast cancer activity, and drug-likeness.

The use of *in-silico* approaches has become an essential tool in cancer drug discovery. These methods have acceler-ated the screening of bioactive molecules and early assessment of their drug-likeness, physicochemical properties, phar-macokinetic behaviour, and potential toxicity, reducing the long timelines, labour, and high costs traditionally associated with experimental drug development [29]. Notably, *in-silico* approaches such as molecular docking, molecular dynamics simulations, and ADMET profiling have been heavily used and are essential tools in screening potential drug candidates [29,30]. A recent study using similar in silico techniques identified a promising human epidermal growth factor receptor 2 kinase inhibitor through virtual screening, and subsequent experimental assays confirmed its micromolar potency [31], showing that robust *in-silico* approaches can be successfully translated into laboratory validation with the potential to inform future clinical development. Therefore, we employed an in silico screening strategy to evaluate 6 selected orga-nosulfur compounds derived from garlic against key breast cancer molecular targets (Bcl-2, XIAP-BIR2, CDK2, CDK6, topoisomerases I and II, VEGFR2, and G-quadruplex DNA) to evaluate their drug-likeness, binding interactions, and potential as novel therapeutic candidates.

## Methods

### Experimental design

This study aimed to assess the multi-target inhibitory potential and drug-likeness of six garlic-derived organosulfur com-pounds against key breast cancer-related molecular targets using in silico approaches. The study was designed to: (i) retrieve and optimize the two-dimensional (2D) structures of selected compounds; (ii) perform molecular docking against relevant breast cancer targets; (iii) carry out post-docking energy minimization; (iv) analyze protein-ligand interactions; and (v) evaluate physicochemical, pharmacokinetic, and toxicity profiles of the compounds. The computational pipeline incorporated validated tools and reproducible parameters to ensure reliable predictions of binding affinity, drug-likeness, and ADMET (absorption, distribution, metabolism, excretion, and toxicity) characteristics.

### Structure of the investigational compounds

The Structure Data File (SDF) containing the 2D structures of six garlic-derived compounds (Fig 1) were retrieved from the PubChem database (https://pubchem.ncbi.nlm.nih.gov): cpd1 (PubChem CID: 9881148), cpd2 (CID: 61926), cpd3 (CID: 16590), cpd4 (CID: 11617), cpd5 (CID: 16315), and cpd6 (CID: 9793905). Geometry optimization was carried out using the MMFF94 force field and steepest descent (maximum 600 steps) algorithm with an RMS gradient convergence criterion of $1 \times 10^{-7}$ after converting the SDF files to MOL2 formats considering the physiological pH of 7.4. File conver-sion and energy minimization were done using the OpenBabel (version 2.4.1) software. PDB files were later prepared in

**Fig 1. Two-dimensional chemical structures of Allium *sativum L.*-derived compounds investigated for their inhibitory potential and physico-chemical properties.**

AutoDockTools software (version 1.5.7) by adding all hydrogen atoms, computing Gasteiger charges, merging the non-polar hydrogens with their bonded carbons and maintaining default torsional angles before exported as Protein Data Bank, Partial Charge, & Atom Type (PDBQT) files for molecular docking.

## Molecular docking and interaction analysis

Eight human molecular targets relevant to breast cancer cell survival, proliferation, and apoptosis in PDB format were selected: CDK-2 (PDB ID: 1DI8, resolution = 2.20 Å) [32], CDK-6 (1XO2, resolution = 2.90 Å) [33], Topoisomerase I (1T8I, resolution = 3.00 Å) [34], Topoisomerase II (1ZXM, resolution = 1.87 Å) [35], G-Quadruplex (1L1H, resolution = 1.75 Å) [36], Bcl-2 (2O2F, resolution not calculated) [37], VEGFR-2 (2OH4, resolution = 2.05 Å) [38], and XIAP-Bir2 (4KJU, resolution = 1.60 Å) [39]. Structures were obtained from the RCSB Protein Data Bank (www.rcsb.org) and prepared by removing heteroatoms (water, ions), adding polar hydrogens, and assigning Kollman charges. The structures were saved as a PDBQT file for docking. Docking was conducted using AutoDock Vina (v1.1.2) with an energy range of 4 and exhaustiveness of 10. Binding sites were defined based on prior studies and co-crystallized ligand positions [40]. The docking grid parameters for all target proteins, including center coordinates, grid dimensions, and spacing, are summarized in Supplementary Table 1. Binding affinity (ΔG) and inhibition constant (Ki) were recorded for each ligand. Ki values were calculated using the equation $Ki = \exp(\Delta G/RT)$, where $R = 1.98$ calK$^{-1}$mol$^1$ and $T = 298.15$ K. PyMol (The PyMOL Molecular Graphics System, Version 3.0 Schrödinger, LLC) and Ligplot+ tool (v.2.3.1) were used to analyze binding poses and interactions of top-ranked ligand-protein complexes. To validate docking accuracy, re-docking of co-crystallized ligands was performed. Compounds with the lowest binding free energy and relevant interactions were selected for further analysis.

## Post-docking energy minimization

Following molecular docking simulations, the protein-ligand complex was subjected to energy minimization to relax the structure and assess binding stability, selecting the best-scoring pose from the docking output. The docked ligand and protein files, initially in PDBQT format, were converted to PDB format using OpenBabel (version 2.4.1) and merged into a single complex file. Energy minimization was performed using OpenBabel's obminimize tool with the Universal Force Field, suitable for diverse atom types, employing 2000 steps and a convergence criterion of $1 \times 10^{-4}$ kcal/mol. Stability was evaluated by calculating the total energy of the complex before and after minimizing using OpenBabel's obenergy tool, with a decrease in energy indicating successful relaxation. Structural stability was further assessed using PyMOL (The PyMOL Molecular Graphics System, Version 3.0 Schrödinger, LLC) by aligning the original and minimized complexes and computing the root-mean-square deviation (RMSD), where an RMSD below 2 Å suggested a stable binding mode.

## Physicochemical and pharmacokinetic analysis

To predict oral bioavailability, drug-likeness, and toxicity, Molinspiration (https://www.molinspiration.com), DataWarrior (v5.5.0), SwissADME (https://www.swissadme.ch), and PreADMET (https://preadmet.webservice.bmdrc.org) tools were used. Parameters assessed included Lipinski's rule of five, polar surface area (PSA), aqueous solubility (cLogS), number of rotatable bonds, and toxicity risks (mutagenicity, tumorigenicity, reproductive effects, and irritation potential). SwissADME and PreADMET also predicted blood-brain barrier penetration, human intestinal absorption, cytochrome P450 inhibition, and plasma protein binding. The potential for P-glycoprotein inhibition was also assessed due to its relevance in drug-drug interaction risks.

## Statistical analysis

This computational study did not involve experimental replicates or inferential statistical analysis. However, reproducibility was ensured through consistent docking parameters and validation via redocking. Quantitative outputs such as ΔG and

Ki were generated using AutoDock Vina and calculated using standard thermodynamic equations. Toxicity and ADMET predictions were reported as categorical outputs from respective software. All data are available within the manuscript and supplemental data and source databases cited. All optimized ligand and receptor structures generated in this study have been deposited in Zenodo and can be accessed at https://doi.org/10.5281/zenodo.17804558.

## Results

### Molecular docking outcomes for lead investigational compounds

To validate the molecular docking protocol before docking the investigational ligands, co-crystallized ligands (LIO, FSE, DTQ, PYN, EDH, ANP, GIG, and 1RH) were re-docked into their respective receptor binding sites (Bcl2, CDK-6, CDK-2, G-Quadruplex, Topoisomerase I, Topoisomerase II, VEGFR-2, and XIAP-Bir2) as shown in Fig 2. The root mean square deviation (RMSD) between the docked and original crystal ligand conformations was calculated, with a cutoff of 2.0 Å for docking reliability. Most co-crystallized ligands yielded RMSD values below this threshold, confirming the method's reproducibility (Fig 3). Notably, ANP and GIG exceeded the 2.0 Å threshold, likely due to their inherent conformational flexibility. This validation approach is consistent with established molecular docking protocols and thereby supports the robustness of subsequent ligand-receptor interaction analyses.

Each investigational compound (Fig 1) and co-crystal ligand (Fig 2) were docked against selected macromolecular targets, and their binding energies (ΔG, kcal/mol) and inhibition constants (Ki, μM) were calculated (Table 1). Compound 1 (cpd1) (Z)-1-(prop-2-enyldisulfanyl)-3-prop-2-enylsulfinylprop-1-ene, commonly known as Z-ajoene, emerged as the most potent inhibitor of Bcl-2, CDK-2, and Topoisomerase II. It showed the strongest binding to Bcl-2 with a binding energy of −4.5 kcal/mol and a Ki of 499.05 μM, interacting mainly via hydrophobic interactions with Phe101(A), Phe150(A), Tyr105(A), Ala146(A), Val130(A), Asp108(A), Phe109(A) and Leu134(A) (Fig 4a). For CDK-2, cpd1 demonstrated a binding energy of −5.1 kcal/mol and Ki of 181.09 μM, forming a hydrogen bond (length = 3.03 Å, angle = 164.8°) with Lys33(A), and hydrophobic interactions with Phe80(A), Ile10(A), Val18(A), Asp145(A), Val64(A), Ala144(A), Phe82(A), Leu83(A), Ala31(A) and Leu134(A) (Fig 4b). Against Topoisomerase II, it achieved a binding energy of −5.4 kcal/mol and Ki of 109.09 μM, interacting via hydrogen bonds with Asn150(A) (length = 2.89 Å, angle = 109.0 °) and Ser149(A) (length = 2.80 Å, angle = 175 °), and hydrophobic interaction with Ser148(A), Ile141(A), Phe142(A), Thr215(A), Asn95(A) and Asn91(A) (Fig 4f). While cpd1 showed favorable binding to all three targets, co-crystal ligands (DTG, LIO, and ANP) demonstrated stronger cooperative binding to their respective targets (CDK-2, Bcl-2, and Topoisomerase II).

S-allyl-L-cysteine (cpd6) was identified as a potent inhibitor of five out of the eight molecular targets: G-quadruplex, Topoisomerase I, VEGFR2, XIAP-Bir2, and CDK-6, with binding free energies of −4.9, −4.8, −4.7, −4.7, and −5.0 kcal/mol, and Ki values of 253.89 μM, 300.62 μM, 355.96 μM, 355.96 μM, and 214.42 μM, respectively. Cpd6 interacted with the G-quadruplex via hydrogen bonds with DG2007(B) (length = 3.28 Å, angle = 107.7 °), DT2006(B) (length = 3.13 Å, angle = 139.7 °), DT2005(B) (length = 2.81 Å, angle = 113.5 °), and DT2008(B) (length = 3.05 Å, angle = 170 °), and hydrophobic interaction with DG2009(B) and DG1012(A) (Fig 4d). For Topoisomerase I, an enzyme that cleaves and re-ligates one strand of DNA during relaxation [41], cpd6 formed hydrogen bonds with Pro212(A) (length = 2.87 Å, angle = 98.9 °), Tyr211(A) (3.01 Å, angle = 124.2 °), Gly214(A) (length = 3.05 Å, angle = 146.5 °), Lys216(A) (length = 3.17 Å, angle = 120.5 °) and Ile215(A) (length = 2.97 Å, angle = 116.5 °) along with hydrophobic interactions with Glu213(A), Gln442(A), Lys439(A), Ile435(A), Glu438(A) and Arg434(A) (Fig 4e). In VEGFR2, cpd6 bonded with Glu883(A) (length = 2.91 Å, angle = 126.1 °) and Asp1044(A) (length = 3.19 Å, angle = 126.6 °) through hydrogen bonding and showed hydrophobic interactions with Val846(A), Lys866(A), Val914(A), Cys1043(A), Leu1033(A) and Phe1045(A) (Fig 4g). Against XIAP-Bir2, it formed hydrogen bonds with Phe228(A) (length = 3.18 Å, angle = 157.2 °) and Asn226(A) (length = 3.10 Å, angle = 135.5 °), along with hydrophobic interactions involving Cys227(A), Cys203(A), Thr152(A), Ile153(A), Lys208(C) and Gly205(C) (Fig 4h). Similarly, in CDK-6, known for regulating the G1 to S phase transition through cyclin D binding [11], cpd6 interacted via hydrogen bonds with Gln149(B) (length = 2.99 Å, angle = 150.6 °) and Asp104(B) (length = 3.08 Å, angle = 100.0 °), and

DTQ:4-[3-Hydroxyanilino]-6,7-dimethoxyquinazoline

FSE:3,7,3',4'-Tetrahydroxyflavone

EDH:4Ethyl-4-hydroxy-1,12-dihydro-4H-2-oxa-6
,12A-diaza-dibenzo[B,H]flourene-3,13-dione

ANP: Phosphoaminophosphonic acid-adenylate ester

PYN:3-Pyrolidin-1-yl-N-[6-(3-pyrrolidin-1
-yl-propionlylamino)-acridin-3-yl]-propiona
mide

LIO:4-(4-Benzyl-4-methoxypiperidin-1-yl)-N-
[(4-{[1,1-dimethyl-2-(phenylthio)ethyl]amino}
-3-nitrophenyl)sulfonyl]benzamide

GIG:Methyl(5-{4-[({[2-flouro-5(triflouromethyl)
phenyl]aminocarnonyl)amino]phenoxy}-1H-Benz
imidazol-2-yl)carbamate

1RH:N-{(3S)-5-(aminobenzoyl)-1-[(2-methoxyna
phthalen-1-yl)methyl]-2-oxy-2,3,4,5-tetrahydro-1
H-1,5-benzodiazepin-3-yl}-N~2~-methyl-l-alanina
mide

**Fig 2. Co-crystallized ligands of the selected macromolecular targets.** The ligand-receptor pairs include CDK2/DTQ, CDK6/FSE, Topoisomerase I/EDH, Topoisomerase II/ANP, G-Quadruplex/PYN, Bcl-2/LIO, and XIAP-Bir2/1RH. These ligands are established inhibitors of their respective targets and were used to validate docking protocols.

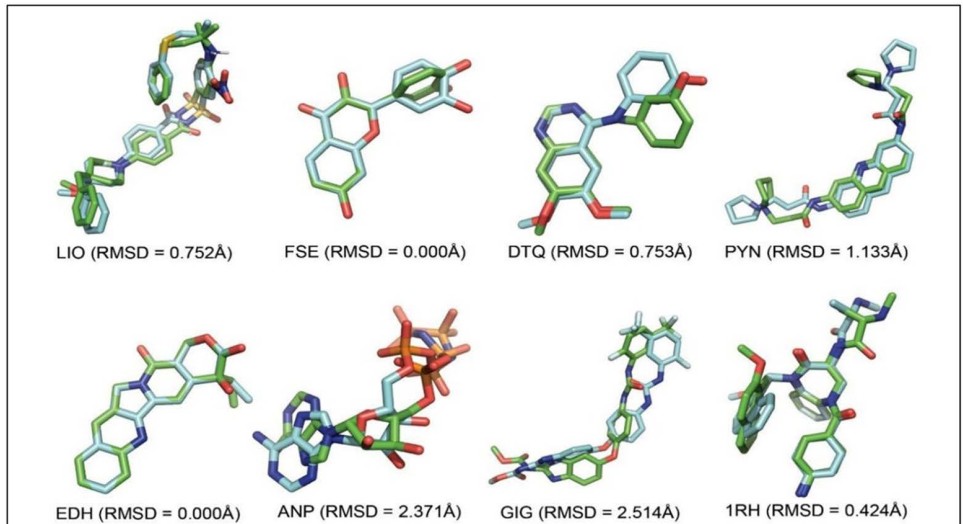

**Fig 3. Conformational clustering of co-crystal ligands used to validate molecular docking parameters.** The RMSD was calculated between the original co-crystal ligand positions (green-colored aromatic carbons) and their docked poses (cyan-colored aromatic carbons). A cut-off RMSD value of 2.0 Å was applied to identify closely related conformers, indicating acceptable docking accuracy.

**Table 1. Binding energy (ΔG, kcal/mol) and inhibition constant (Ki, µM) of organosulfur compounds from Allium sativum L. docked against selected macromolecular targets.**

| Target | CDK-6 | | CDK-2 | | Bcl-2 | | VEGFR-2 | | XIAP-Bir2 | | G-Quadruplex | | Topoisomerase I | | Topoisomerase II | |
|---|---|---|---|---|---|---|---|---|---|---|---|---|---|---|---|---|
| Ligands | ΔG | Ki | ΔG | Ki | ΔG | Ki | ΔG | Ki | ΔG | Ki | ΔG | Ki | ΔG | Ki | ΔG | Ki |
| cd1 | −4.3 | 699.67 | **−5.1** | **181.09** | **−4.5** | **499.05** | −4.4 | 590.91 | −4.5 | 499.05 | −4.1 | 980.94 | −4.1 | 980.94 | **−5.4** | **109.09** |
| cd2 | −3.7 | 1928.16 | −3.6 | 2283.06 | −3.6 | 2283.06 | −3.4 | 3200.86 | −3.1 | 5313.61 | −3.1 | 5313.61 | −3 | 6291.65 | −3.3 | 3790.01 |
| cd3 | −4.2 | 828.46 | −4.1 | 980.94 | −4 | 1161.5 | −4.1 | 980.94 | −3.4 | 3200.86 | −3.3 | 3790.01 | −3.8 | 1628.43 | −3.6 | 2283.06 |
| cd4 | −4 | 1161.5 | −3.9 | 1375.29 | −4 | 1161.5 | −3.7 | 1928.16 | −3.5 | 2703.28 | −3.3 | 3790.01 | −3.5 | 2703.28 | −3.9 | 1375.29 |
| cd5 | −4.3 | 699.67 | −4.4 | 590.91 | −4.2 | 828.46 | −3.9 | 1375.29 | −3.6 | 2283.06 | −3.3 | 3790.01 | −3.5 | 2703.28 | −4.1 | 980.94 |
| cd6 | **−5** | **214.42** | −4.7 | 355.96 | −4.2 | 828.46 | **−4.7** | **355.96** | **−4.7** | **355.96** | **−5.2** | **148.97** | **−4.8** | **300.62** | −5.3 | 129.17 |
| Co-crystal ligands | −7.6 | 2.65 | −8.5 | 0.58 | −10.3 | 0.03 | −10.9 | 0.01 | −7.5 | 3.14 | −9.5 | 0.11 | −8.2 | 0.96 | −10.5 | 0.2 |

Bold values indicate the compound with the highest binding affinity (lowest ΔG/ Ki) for each target.

hydrophobic interactions with Gln103(B), Phe98(B), Ala41(B), Ala162(B), Leu152(B) and Val77(B) (Fig 4c). Although cpd6 was less potent than co-crystallized ligands (PYN, EDH, GIG, FSE, and 1RH), it demonstrated broad target engagement across multiple key molecular targets (G-quadruplex, topoisomerase I, VEGFR2, XIAP-Bir2, and CDK-6).

## Post-docking energy minimization of best-docked compound-receptor complex

Post-docking energy minimization was aimed at refining the predicted protein-ligand complex by reducing steric clashes, optimizing atomic geometries, and arriving at a more stable conformation that better represents physiological conditions. After post-docking energy minimization, all the best-docked-compound-ligand complexes showed stability with lower energy compared to pre-minimized structures in addition to the retention of similar pose (RMSD's less than 2.0 Å) of the

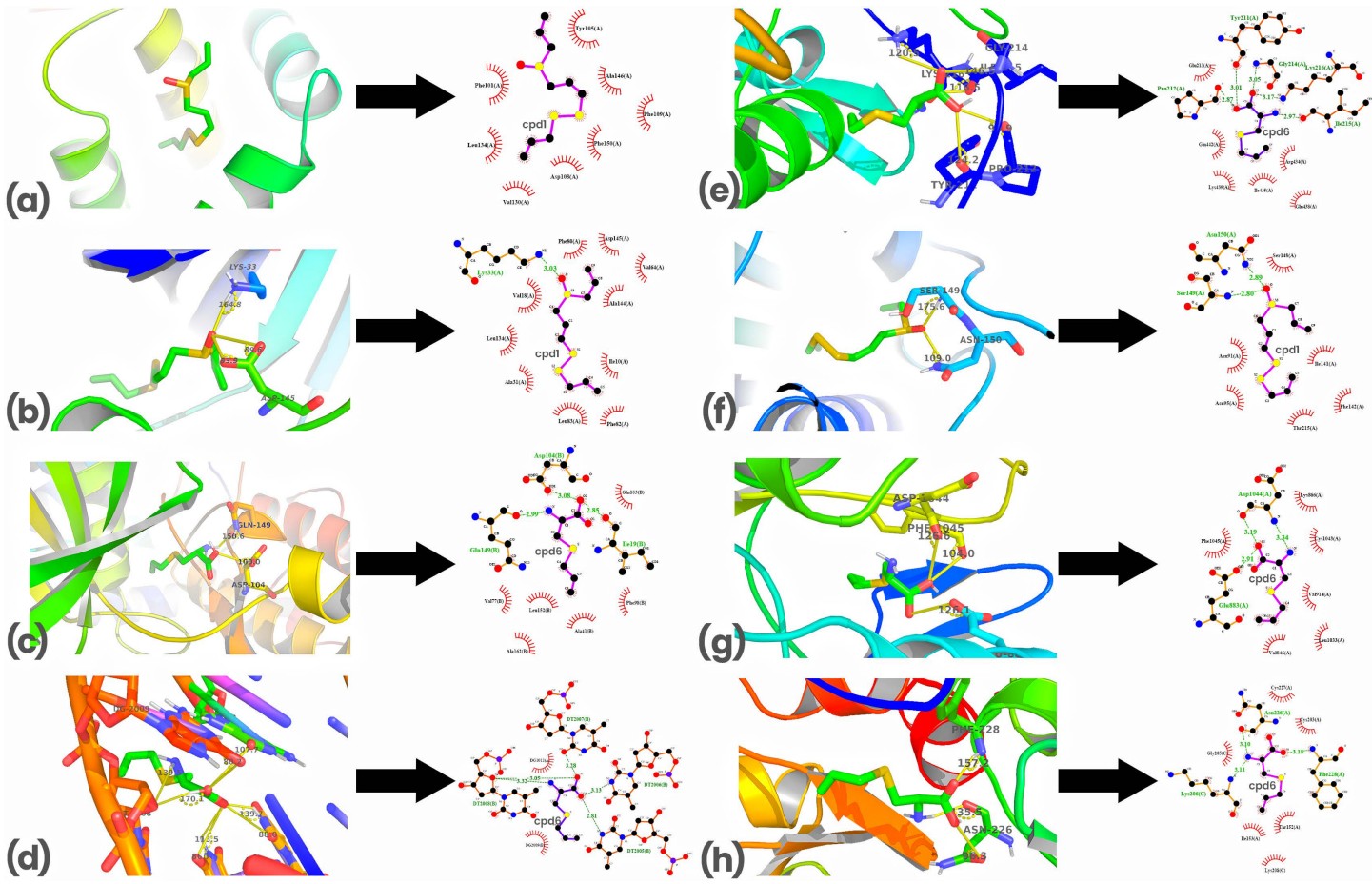

**Fig 4. Binding modes and LigPlot+ molecular interactions of the best-docked investigational ligands with their molecular targets.** Bond lengths shown in Å and hydrogen bonding angles between the ligands and key amino acid or nucleotide residues of the receptors are annotated on the LigPlot+ and binding mode images respectively. Z-ajoene (cpd1) showed notable binding with Bcl-2 (Panel 4a), CDK2 (Panel 4b), and Topoisomerase II (Panel 4f) through a combination of hydrogen bonding and hydrophobic interactions. S-allyl-L-cysteine (cpd6) demonstrated strong interactions with G-Quadruplex (Panel 4d), Topoisomerase I (Fig 4e), VEGFR2 (Panel 4g), XIAP-Bir2 (Panel 4h), and CDK-6 (Panel 4c), involving hydrogen bonds and multiple hydrophobic contacts. Specific interacting residues and bond types are highlighted in each panel (Panel 4a-4h).

ligands after the energy minimization. Among the complexes, Cpd1/VEGR2 complex showed a major reduction in energy after minimization compared to other complexes (Fig 5a). Only Cpd6/G-Quadruplex complex had an RMSD value of ~1.0 Å, making it the complex with the highest deviation from the pre-minimized pose but lesser than the cut-off value of 2.0 Å. All other complexes had RMSD value approximately <0.6 Å, indicating good similarity between unminimized and minimized poses (Fig 5b-j).

## Physicochemical properties and drug-likeness of the investigational compounds

Given the high attrition rate of drug candidates during later stages of development [42], we assessed the physicochemical and ADMET properties of the investigational compounds using open-source computational tools. According to Molinspiration analysis (Table 2), all six compounds complied with Lipinski's Rule of Five (ROF), a widely used guideline for predicting oral bioavailability. The ROF criteria include molecular weight (MW) < 500, lipophilicity (logP) < 5, number of hydrogen bond donors < 5, and hydrogen bond acceptors < 10 [43]. This suggests that the compounds may possess favorable

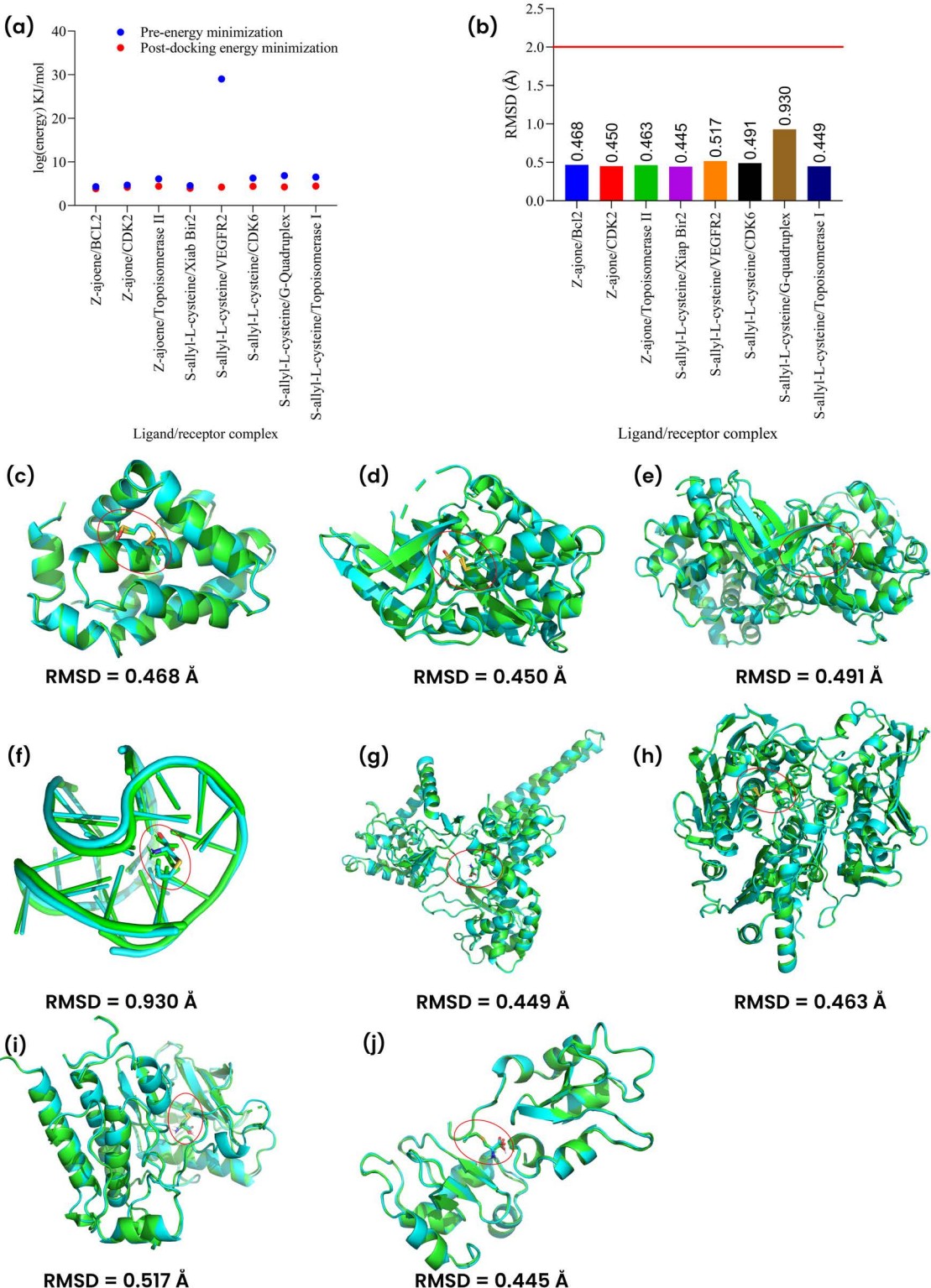

**Fig 5. Post-docking energy minimization using OpenBabel tool.** Energy values showed that ligand-receptor complexes have lower energy post-docking minimization (Panel 5a). An RMSD value of 2.0 Å was considered as cut-off for ligands that remain in similar poses after aligning

pre-energy minimization complexes (green colored carbon backbone) with post-docking energy minimization complexes (cyan colored carbon backbone) (Panel 5b-j). Complexes: Cpd1/Bcl2 (Panel 5c), Cpd1/Cdk2 (Panel 5d), Cpd6/Cdk6 (Panel 5e), Cpd6/G-Quadruplex (Panel 5f), Cpd6/Topoisomerase II (Panel 5g), Cpd1/Topoisomerase I (Fig 5h), Cpd6/VEGFR2 (Panel 5i) and Cpd6/XIAP Bir2 (Panel 5j).

**Table 2. Predicted physicochemical properties of the investigational bioactive compounds calculated using the Molinspiration web server.**

| Compound | Molecular weight | Partition coefficient between n-Octanol and water (logP) | Number of hydrogen bond acceptors | Number of hydrogen bond donors | number of rules violated |
|---|---|---|---|---|---|
| cpd1 | 234.41 | 1.8 | 1 | 0 | 0 |
| cpd2 | 152.31 | 2.48 | 0 | 0 | 0 |
| cpd3 | 146.28 | 2.63 | 0 | 0 | 0 |
| cpd4 | 114.21 | 2.13 | 0 | 0 | 0 |
| cpd5 | 178.35 | 3.13 | 0 | 0 | 0 |
| cpd6 | 161.23 | −1.86 | 3 | 3 | 0 |

properties for membrane permeability, absorption, and oral administration. To further evaluate drug-likeness, we examined additional ADMET-related parameters using the DataWarrior program (version 5.5.0), including aqueous solubility at 25 °C and pH 7.5 (cLogS > −4.0), polar surface area (PSA < 140 Å²), and number of rotatable bonds (RTB < 10) (Table 3). Toxicological risk factors such as mutagenicity, tumorigenicity, reproductive toxicity, irritation potential, and molecular complexity were also assessed. The DataWarrior program calculations (Table 3) indicate that all six bioactive compounds possess good solubility and oral bioavailability, as none violate the cutoff values for cLogS, PSA, and RTB. Most of the compounds exhibited no mutagenic potential, except for compound 4 (diallyl sulfide), which showed a potential mutagenic effect. Furthermore, all compounds were predicted to be non-tumorigenic, non-irritant, and non-reproductive toxicants.

Additional physicochemical properties related to oral bioavailability were assessed using the Veber and Egan rules. According to the Veber rule, compounds with ≤10 RTB and a PSA ≤ 140 Å² are considered to have good oral bioavailability [44]. The Egan rule considers compounds with LogP ≤ 5.88 and PSA ≤ 131.6 Å² as potentially orally bioavailable [45]. Molecular complexity, expressed as the ratio of sp³-hybridized carbons to total carbon count [46], was also considered. Results from SwissADME (Table 4) show that all the compounds comply with both the Veber and Egan criteria and demonstrate good bioavailability scores. The high bioavailability scores observed may also be attributed to the relatively low molecular complexity of the compounds.

PreADME was used to evaluate key pharmacokinetic properties, including blood-brain barrier (BBB) penetration, human intestinal absorption (HIA), P-glycoprotein (P-gp) inhibition, and plasma protein binding, based on established

**Table 3. Physicochemical properties of the investigational compounds calculated using DataWarrior software.**

| Compound | cLogSa | PSA(Å²)b | Mutagenic | Tumorigenic | Reproductive Effect | Irritant | RTBc | Mcpx.d |
|---|---|---|---|---|---|---|---|---|
| cpd1 | −2.446 | 86.88 | none | none | none | none | 8 | 0.37 |
| cpd2 | −2.864 | 75.9 | none | none | none | none | 4 | 0.37 |
| cpd3 | −2.706 | 50.6 | none | none | none | none | 5 | 0.46 |
| cpd4 | −2.009 | 25.3 | high | none | none | none | 4 | 0.37 |
| cpd5 | −3.305 | 75.9 | none | none | none | none | 6 | 0.35 |
| cpd6 | −1.222 | 88.62 | none | none | none | none | 5 | 0.60 |

[a]aqueous solubility at 25°C and pH 7.4 [b] polar surface area (sum of surface area of nitrogen and oxygen plus hydrogen attached to the heteroatoms)
[c]number of rotatable bonds [d] molecular complexity

**Table 4. Oral bioavailability of the selected compounds based on Veber and Egan Rules as predicted by the SwissADME server.**

| Compound | VEBER | EGAN | Bioavailability Score |
|---|---|---|---|
| cpd1 | Pass | Pass | 0.55 |
| cpd2 | Pass | Pass | 0.55 |
| cpd3 | Pass | Pass | 0.55 |
| cpd4 | Pass | Pass | 0.55 |
| cpd5 | Pass | Pass | 0.55 |
| cpd6 | Pass | Pass | 0.55 |

criteria (Table 5). In addition, SwissADME was employed to assess the inhibitory potential of the compounds on cytochrome P450 isoenzymes CYP2C19 and CYP2C9 (Table 5). Plasma protein binding can be either reversible or irreversible, and this property is commonly used in clinical settings to estimate or determine the therapeutic dose of a drug [47,48]. Compounds are considered strongly bound to plasma proteins if they exhibit a binding score of >90% and weakly bound if <90%. According to the PreADME results, all six compounds demonstrated weak binding to plasma proteins, except diallyl disulfide (cpd 3), which showed a comparatively stronger binding affinity. This finding suggests that the in vivo bioavailability of diallyl disulfide may be lower due to its higher plasma protein binding.

Prediction of HIA is a critical step in the design, optimization, and selection of orally administered drugs [49]. HIA was evaluated based on the following criteria: compounds with HIA values between 0–20% are considered poorly absorbed, those between 20–70% moderately absorbed, and those between 70–100% highly absorbed [40]. Based on this assessment, all the compounds investigated demonstrated high HIA values, with diallyl sulfide showing the highest absorption (100%) and compound 6 the lowest (87.97%) (Table 5). Regarding P-gp inhibition, most of the compounds (cp2, cp3, cp4, and cp5) were predicted to be potential P-gp inhibitors, except for Z-ajoene and S-allyl-L-cysteine, which exhibited no inhibitory activity. P-glycoprotein is a member of the ATP-binding cassette superfamily of membrane transporters and plays a key role in the efflux of various xenobiotics, including drugs [50]. It is also a major component of the BBB. Inhibition of P-gp may increase the risk of drug-drug interactions and potential toxicity in vivo [50].

The BBB forms a highly selective permeability barrier between the central nervous system (CNS) and the systemic circulation. For CNS-targeting drugs, high BBB permeability is desirable, while for drugs targeting peripheral organs, low BBB penetration is preferred [51]. Based on the following classification, high CNS penetration (BB value > 2.0), moderate

**Table 5. Absorption, distribution, elimination, and toxicity profiles of the bioactive compounds assessed using PreADME; CYP2C19 and CYP2C9 inhibition predicted by SwissADME.**

| Compound | HIA[a] | BBB[b] | Pgb_I[c] | PPB[d] | CYP2C19 inhibitor | CYP2C9 inhibitor |
|---|---|---|---|---|---|---|
| cpd1 | 99.31 | 1.04 | No | 70.8 | No | Yes |
| cpd2 | 98.63 | 1.83 | Yes | 55.47 | No | No |
| cpd3 | 98.12 | 1.37 | Yes | 98.04 | No | No |
| cpd4 | 100 | 0.8 | Yes | 78.05 | No | No |
| cpd5 | 99 | 2.28 | Yes | 55.56 | No | No |
| cpd6 | 87.97 | 0.22 | No | 11.67 | No | No |

[a]human intestinal absorption

[b]blood-brain barrier penetration

[c]p-glycoprotein inhibition

[d]plasma protein binding

penetration (0.1 < BB ≤ 2.0), and low penetration (BB < 0.1), most compounds were predicted to have moderate BBB permeability, except cpd 5, which showed high absorption potential (Table 5). According to the SwissADME results (Table 5), most of the investigational compounds are not predicted to inhibit CYP2C19 or CYP2C9, except Z-ajoene, which shows potential inhibition of CYP2C9.

## Discussion

This study employed computational approaches to evaluate the drug-likeness and molecular interactions of garlic-derived organosulfur compounds to reveal their potential to disrupt critical multi-targets involved in breast cancer progression and resistance. Our results pointed to Z-ajoene and S-allyl-L-cysteine as the most promising candidates, showing good binding affinities to key cancer-related proteins such as Bcl-2, CDK-2, and topoisomerase II. In addition to similar binding pose retention and reduced energy post-docking minimization, the favorable ADMET profiles and adherence to Lipinski's ROF suggest these compounds may possess desirable pharmacokinetic properties, reinforcing their potential as lead compounds for further development in breast cancer therapy.

Natural products make up about 60% of all cancer drugs and have been shown to slow breast cancer growth through alteration of key pathways and help overcome multidrug resistance by making cancer cells more responsive to treatment [52–54]. Garlic is known for its anticancer properties, largely attributed to its major organosulfur compounds, which have shown notable therapeutic efficacy in preclinical studies [26, 28, 55,56]. The importance of screening garlic-derived compounds for breast cancer therapy lies in their structural diversity and low toxicity, offering a safer alternative to many synthetic chemotherapeutics that often cause off-target effects [57]. For instance, key garlic organosulfur compounds such as diallyl trisulfide and ajoene have shown promising activity, with diallyl trisulfide modulating redox pathways in triple-negative breast cancer cells, and ajoene disrupting protein folding in the endoplasmic reticulum, a known vulnerability in drug-resistant cancers [58]. Expanding such investigations may reveal compounds with synergistic potential when combined with existing therapies, similar to the success achieved with plant-derived agents like paclitaxel [59].

The molecular docking and ADMET profiling used in this study to assess the therapeutic potential of six garlic-derived organosulfur compounds against key breast cancer targets revealed distinct binding affinities and pharmacokinetic properties, with notable variations in target specificity and multi-target engagement. Z-ajoene exhibited strong binding affinities to multiple breast cancer targets, particularly Bcl-2 and topoisomerase II. This dual interaction suggests potential for both apoptosis induction and enhancement of DNA damage, a strategy that could help overcome resistance mechanisms involving apoptotic evasion [60]. Specifically, Z-ajoene's interaction with Bcl-2 involving hydrophobic contacts with Phe101, Phe150 and Tyr105 may suggest a disruption in the protein's anti-apoptotic function, a key factor in cancer cell survival [61,62]. In addition, Z-ajoene binds to CDK-2, a key regulator of the G1-to-S phase transition, and this binding mode is similar to how FDA-approved CDK4/6 inhibitors like palbociclib interact with their targets, suggesting that Z-ajoene may exert comparable cell cycle inhibitory effects [63]. Notably, Z-ajoene's interaction with topoisomerase II mimics that of etoposide, though with lower affinity, indicating potential for future structural optimization [64].

S-allyl-L-cysteine, although less potent than co-crystallized inhibitors such as PYN, EDH, GIG, FSE, and 1RH, demonstrated broad target engagement by binding to G-quadruplex DNA, topoisomerase I, VEGFR2, XIAP-BIR2, and CDK-6. Importantly, a direct comparison with the FDA-approved CDK4/6 inhibitor Abemaciclib highlights both the limitations and opportunities of this garlic-derived compound. While Abemaciclib shows much stronger binding to CDK6 (ΔG ≈ −12.12 kcal/mol) [65] compared to S-allyl-L-cysteine (−5 kcal/mol in silico), the latter engages critical residues in the ATP-binding pocket through hydrogen bonding with Gln149 and Asp104, as well as hydrophobic contacts with Gln103, Phe98, Ala41, Ala162, Leu152, and Val77. These interactions suggest that S-allyl-L-cysteine provides a viable structural scaffold for G1/S phase inhibition. Moreover, unlike synthetic inhibitors such as Abemaciclib, S-allyl-L-cysteine offers potential advantages in biocompatibility and reduced toxicity risk. With rational optimization, such as introducing π-π stacking moieties or enhancing hydrogen bond donors to mimic Abemaciclib's interactions with Val101 and His100 [65,66], its affinity and

selectivity could be significantly improved. This broad-spectrum activity supports a polypharmacological approach, where multi-target engagement may reduce the risk of resistance development by breast cancer cells [67]. Notably, its stabilization of G-quadruplex structures, mediated by interactions with DG1012 and DT2008, suggests potential suppression of oncogenes such as c-MYC, a strategy employed by other quadruplex-binding agents like quarfloxin [68]. Furthermore, its binding to VEGFR2 indicates anti-angiogenic potential, paralleling the mechanism of clinical inhibitors such as cediranib, although affinity optimization would be necessary to improve its efficacy [69].

The efficacy of the compounds investigated, particularly diallyl disulfide and diallyl trisulfide has been established in liver, stomach, bone, and lung cancers, nonetheless, a detailed understanding of their multi-target profiles and site-specific binding remains fragmented in the context of breast cancer [70–75]. Our research bridges this gap, employing computational modeling to elucidate the molecular interactions between these organosulfur compounds and key protein targets. Our findings regarding Bcl-2 align with previous experimental reports suggesting that the compounds, particularly, diallyl trisulfide, diallyl disulfide and diallyl sulfide could induce apoptosis in breast cancer. However, comparative docking scores indicate that Z-ajoene has a higher binding affinity than the other investigated compounds, which may explain the higher potency observed in Z-ajoene-treated cell lines [23]. Notably, the affinity interaction identified between these ligands and the XIAP-Bir2 domain and the G-Quadruplex represents a significant expansion of the known medicinal chemistry of garlic derivatives. While most existing literature focuses on Bcl-2 and other targets inhibition, our data positions VEGFR-2, Topoisomerase I and Topoisomerase II as equally viable primary targets.

While the investigational compounds showed moderate binding affinities (Ki ~ $10^2$-$10^3$ μM) compared to established chemotherapeutics like doxorubicin (Ki ≈ 1 μM for topoisomerase II) [76,77], their strong drug-likeness and pharmacokinetic profiles suggest significant potential for further development. In addition, the moderate binding affinities demonstrated by the compound is therapeutically important in rational drug design such that the weaker interaction with the receptors implies weaker binding to non-target proteins which help minimize side effects [78]. Compounds with high binding affinity could sometimes be difficult to develop into a drug as they might have poor solubility or bioavailability [79]. The moderate bindings could allow for greater flexibility in the design of other drug-like properties of these investigational compounds [80]. Further, all the compounds met key criteria for oral bioavailability and demonstrated high predicted intestinal absorption with minimal toxicity risks. Moderate plasma protein binding may enhance bioavailability [81], however, potential inhibition of P-glycoprotein by compounds cpd2, cpd3, cpd4, and cpd5, and CYP2C9 inhibition by cpd1 (Z-ajoene), raises concerns about drug-drug interactions. Inhibition of P-gp could increase the absorption and tissue distribution of substrate drugs, while CYP2C9 inhibition by Z-ajoene may reduce the metabolism and clearance of co-administered drugs, potentially leading to elevated plasma levels and increased risk of toxicity or therapeutic failure [82,83]. This highlights the need for careful monitoring and possible dose adjustments when used alongside medications metabolized by CYP2C9 or transported by P-gp. Nonetheless, these properties support their promise as scaffolds for optimization in breast cancer therapy. The properties of these compounds could be improved by structural optimization using structure activity relationship method.

## Conclusion

This study utilized computational methods to evaluate the multi-target inhibitory potential of six garlic-derived organosulfur compounds. Z-ajoene and S-allyl-L-cysteine emerged as promising candidates, demonstrating relatively strong binding affinities and broad-spectrum activity against key breast cancer biomarkers. These compounds also exhibited favorable drug-likeness and pharmacokinetic profiles. As this study is based solely on in silico analysis, experimental validation in appropriate in vitro and in vivo breast cancer models will be essential to confirm and extend these observations before any therapeutic implications can be established. The moderate binding affinities of the top two compounds further suggest a need for further structural optimization to enhance potency, selectivity, and overall efficacy. This work provides preliminary yet meaningful evidence to support the therapeutic potential of garlic-derived organosulfur compounds for breast cancer.

This study primarily relied on molecular docking and post-docking energy minimization to predict the binding affinities and interactions of garlic-derived organosulfur compounds with key breast cancer targets. While docking provides useful initial insights into potential ligand-target interactions, it does not necessarily mean these outcomes can be translated to clinical context. *In vitro* experimental assays to confirm binding efficacy, cytotoxicity, and pharmacokinetic behavior are essential next steps. The lead compounds: Z-ajoene and S-allyl-L-cysteine identified should undergo structural optimization and preclinical testing to assess their therapeutic potential and safety profiles in breast cancer models.

## Supporting information

**S1 File. Supplemental data.**
(DOCX)

## Author contributions

**Conceptualization:** Courage Siame, Lily Paemka.

**Data curation:** Courage Siame.

**Formal analysis:** Courage Siame.

**Investigation:** Courage Siame.

**Methodology:** Courage Siame.

**Software:** Courage Siame.

**Supervision:** Lily Paemka, Kwabena Owusu Danquah.

**Validation:** Courage Siame, Benedict Ofori.

**Visualization:** Benedict Ofori.

**Writing – original draft:** Courage Siame, Benedict Ofori.

**Writing – review & editing:** Benedict Ofori, Lily Paemka, Kwabena Owusu Danquah.

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
