## [Decision Letter · Decision Letter 0]

8 Aug 2025

PONE-D-25-34728In silico evaluation of garlic-derived organosulfur compounds as multi-target inhibitors of breast cancer biomarkersPLOS ONE

Dear Dr. Danquah,

Thank you for submitting your manuscript to PLOS ONE. After careful consideration, we feel that it has merit but does not fully meet PLOS ONE’s publication criteria as it currently stands. Therefore, we invite you to submit a revised version of the manuscript that addresses the points raised during the review process.

If applicable, we recommend that you deposit your laboratory protocols in protocols.io to enhance the reproducibility of your results. Protocols.io assigns your protocol its own identifier (DOI) so that it can be cited independently in the future. For instructions see: https://journals.plos.org/plosone/s/submission-guidelines#loc-laboratory-protocols. Additionally, PLOS ONE offers an option for publishing peer-reviewed Lab Protocol articles, which describe protocols hosted on protocols.io. Read more information on sharing protocols at . Additionally, PLOS ONE offers an option for publishing peer-reviewed Lab Protocol articles, which describe protocols hosted on protocols.io. Read more information on sharing protocols at https://plos.org/protocols?utm_medium=editorial-email&utm_source=authorletters&utm_campaign=protocols..

We look forward to receiving your revised manuscript.

Kind regards,

Ahmed A. Al-Karmalawy, PhD

Academic Editor

PLOS ONE

Journal Requirements:

Reviewers' comments:

Reviewer's Responses to Questions

**Comments to the Author**

1. Is the manuscript technically sound, and do the data support the conclusions?

Reviewer #1: Yes

Reviewer #2: Yes

2. Has the statistical analysis been performed appropriately and rigorously? 

Reviewer #1: N/A

Reviewer #2: N/A

3. Have the authors made all data underlying the findings in their manuscript fully available?

Reviewer #1: No

Reviewer #2: Yes

4. Is the manuscript presented in an intelligible fashion and written in standard English?

Reviewer #1: Yes

Reviewer #2: Yes

5. Review Comments to the Author

Reviewer #1: 1- Include molecular dynamics (MD) simulations or post-docking energy minimization to assess the stability of protein-ligand complexes over time. This is crucial to support docking results.

2- because of the weak binding affinity the author need to discuss the therapeutic relevance of such moderate affinities in more detail, and consider whether structure-based optimization could improve potency.

3- Inadequate Comparison with Existing Drugs, author need to provide comparative data or references to contextualize how garlic compounds perform relative to standard breast cancer drugs.

4- Conclusion emphasizes therapeutic potential despite no experimental (in vitro/in vivo) validation. Author needs to acknowledge that findings are preliminary and hypothesis-generating, pending experimental confirmation.

5- The manuscript lacks a clear justification for why the 8 specific targets (e.g., CDK2, VEGFR2, G-quadruplex) were chosen. Author needs to justify the clinical relevance of each target in breast cancer pathophysiology and resistance mechanisms.

6- Figures lack resolution, and tables are dense without highlighting top-performing compounds.

7- Certain points (e.g., garlic's anticancer potential) are repeated verbatim. Condense overlapping information to improve focus and flow.

Reviewer #2: The authors of the provided manuscript explored the potential anti-cancer activity of garlic-derived organosulfur compounds targeting several breast cancer-associated biotargets. The manuscript is relevant in the field of drug discovery. Comments are to be addressed as follows:

1. In the molecular modelling simulation section, brief description regarding the topology and binding site description of the five investigated breast cancer-associated biotarget’s. Further, the key reported binding residues should be highlighted as being mentioned important within the current literature.

2. Further, the compounds’ polar interaction patterns with key pocket residues should be annotated in terms of both the bond distances and angles. Hydrogen bonding should be presented within hydrogen bond distances as well as bond angles since hydrogen bond depend on both. Authors should mention the Hydrogen bond angles as well as their distances, since the strength of hydrogen bonding is based on both parameters in a way to ensure the adequacy of optimum hydrogen bonding.

3. The authors should provide MM_GBSA or PBSA energy calculations. It is advised to provide the dissected energy terms of these total energies (ΔGbind Lipo, ΔGbind Solv GB, ΔGbind vDW, ΔGbind Coulomb, and ΔGbind Ligand strain) to further evaluate the nature of interaction (dominant energy potential) based on these different energy terms. This would guide further hit to lead and lead optimization steps.

4. Based on the study results, what are the take-away messages. Authors are advised to highlight the future suggested structural modifications that would improve the hits’ predicted activities based on the computational findings. These insights would be beneficial for guiding future lead optimization and development.

6. PLOS authors have the option to publish the peer review history of their article (what does this mean?). If published, this will include your full peer review and any attached files.). If published, this will include your full peer review and any attached files.

.

Reviewer #1: **Yes**

Reviewer #2: **Yes**

While revising your submission, please upload your figure files to the Preflight Analysis and Conversion Engine (PACE) digital diagnostic tool, https://pacev2.apexcovantage.com/. PACE helps ensure that figures meet PLOS requirements. To use PACE, you must first register as a user. Registration is free. Then, login and navigate to the UPLOAD tab, where you will find detailed instructions on how to use the tool. If you encounter any issues or have any questions when using PACE, please email PLOS at . PACE helps ensure that figures meet PLOS requirements. To use PACE, you must first register as a user. Registration is free. Then, login and navigate to the UPLOAD tab, where you will find detailed instructions on how to use the tool. If you encounter any issues or have any questions when using PACE, please email PLOS at figures@plos.org. Please note that Supporting Information files do not need this step.. Please note that Supporting Information files do not need this step.

---

## [Author Response · Author response to Decision Letter 1]

20 Sep 2025

Dear Dr. Al-Karmalawy,

We appreciate the time and effort you and the reviewers have dedicated to the thorough examination of our manuscript titled ‘In silico evaluation of garlic-derived organosulfur compounds as multi-target inhibitors of breast cancer biomarkers.’ Your constructive feedback has been invaluable in improving the overall quality and clarity of our work. To address your comments, we have carefully reviewed and revised the manuscript to address all concerns raised by the reviewers.

We believe that the manuscript is now suitable for publication in PLOS ONE.

Sincerely,

Kwabena O. Danquah

Corresponding author (on behalf of all authors)

Reviewer #1

Comment 1

Include molecular dynamics (MD) simulations or post-docking energy minimization to assess the stability of protein-ligand complexes over time. This is crucial to support docking results

Response

Thank you for your valuable feedback and for suggesting the inclusion of molecular dynamics simulations or post-docking energy minimization to evaluate the stability of protein-ligand complexes over time. In response, we have performed post-docking energy minimization to assess the stability of the protein-ligand complexes. Details of this additional analysis have been incorporated into the Methodology (lines 134-146), and the corresponding findings are presented in the Results (lines 251-263) and illustrated in Figure 5. Furthermore, these results have been discussed in the Discussion section to reflect their implications.

Comment 2

Because of the weak binding affinity the author need to discuss the therapeutic relevance of such moderate affinities in more detail, and consider whether structure-based optimization could improve potency.

Response

We thank the reviewer for this valuable suggestion. In the original version of the manuscript, we had included a note on the potential for structural optimization to improve the potency of the compounds. In the revised manuscript, we have expanded this discussion to also address the therapeutic relevance of the weak binding affinities observed. This has now been incorporated into the Discussion section (lines 411-414). We believe that this addition provides a more balanced perspective on the translational potential of the compounds and highlights possible avenues for future drug development.

Comment 3

Inadequate Comparison with Existing Drugs, author need to provide comparative data or references to contextualize how garlic compounds perform relative to standard breast cancer drugs.

Response

We thank the reviewer for this important observation and suggestion. In the revised Discussion, we have expanded our comparisons between garlic-derived compounds and FDA-approved inhibitors as well as previously reported in silico studies. Specifically, we now compare S-allyl-L-cysteine’s binding to CDK6 with that of Abemaciclib, an FDA-approved CDK4/6 inhibitor (ΔG ≈ -12.12 kcal/mol vs -5 kcal/mol, respectively), and highlight how S-allyl-L-cysteine engages critical residues in the ATP-binding pocket while offering potential advantages in biocompatibility and reduced toxicity. We also discuss how rational structural modifications could enhance its affinity and selectivity, drawing from published interaction patterns of Abemaciclib. Similarly, Z-ajoene’s binding to CDK-2 and topoisomerase II is contextualized by comparison with palbociclib and etoposide, respectively. These additions provide a clearer framework for interpreting our results relative to clinically relevant standards. The new text can be found in the Discussion section, lines 388-399.

Comment 4

Conclusion emphasizes therapeutic potential despite no experimental (in vitro/in vivo) validation. Author needs to acknowledge that findings are preliminary and hypothesis-generating, pending experimental confirmation.

Response

We thank the reviewer for this valuable feedback. While we had noted in the original submission that the findings are preliminary and require experimental validation, we agree that this point needed to be emphasized more strongly. In the revised manuscript, we have made this limitation explicitly clear in the Abstract and Conclusion, highlighting that our results should be regarded as hypothesis-generating and that further in vitro and in vivo validation will be essential to confirm the therapeutic potential of the identified compounds against breast cancer targets. We believe this addition provides a more balanced and transparent interpretation of the study outcomes.

Comment 5

The manuscript lacks a clear justification for why the 8 specific targets (e.g., CDK2, VEGFR2, G-quadruplex) were chosen. Author needs to justify the clinical relevance of each target in breast cancer pathophysiology and resistance mechanisms

Response

We thank the reviewer for this valuable feedback. While the original manuscript (lines 54–64) provided a general rationale for selecting the eight targets, we agree that a clearer justification was needed. In the revised version, we have expanded the Background section (lines 64-66) to emphasize that these macromolecules are currently targeted by existing breast cancer therapies and to highlight the clinical relevance of each target in breast cancer pathophysiology and resistance. For example, CDKs and topoisomerases are linked to unchecked proliferation and endocrine resistance, Bcl-2 and XIAP mediate apoptosis evasion and chemoresistance, VEGFR2 is central to angiogenesis and tumor progression, and G-quadruplex structures play a role in telomerase regulation and genomic instability. This addition provides an additional layer of information to support the selection of these specific targets and shows their therapeutic importance in drug-resistant breast cancer.

Comment 6

Figures lack resolution, and tables are dense without highlighting top-performing compounds.

Response

We thank the reviewer for this important observation. In response, we have revised the relevant figures and tables to improve clarity and readability. Specifically, in Table 1, the top-performing docked compounds against each macromolecule have been highlighted in bold and a table legend added for easy identification. In addition, all figures have been updated with high-resolution images (≥300 dpi) to ensure better quality in the revised manuscript.

Comment 7

Certain points (e.g., garlic's anticancer potential) are repeated verbatim. Condense overlapping information to improve focus and flow.

Response

We thank the reviewer for this observation. In the revised manuscript, we have carefully reviewed and condensed sections where overlapping information was presented, particularly regarding garlic’s anticancer potential. Redundant statements have been removed or rephrased to improve focus, clarity, and flow, ensuring that each point is presented only once in the most appropriate context.

Reviewer #2

Comment

In the molecular modelling simulation section, brief description regarding the topology and binding site description of the five investigated breast cancer-associated biotarget’s. Further, the key reported binding residues should be highlighted as being mentioned important within the current literature. Further, the compounds’ polar interaction patterns with key pocket residues should be annotated in terms of both the bond distances and angles. Hydrogen bonding should be presented within hydrogen bond distances as well as bond angles since hydrogen bond depend on both. Authors should mention the Hydrogen bond angles as well as their distances, since the strength of hydrogen bonding is based on both parameters in a way to ensure the adequacy of optimum hydrogen bonding.

Response

Thank you for the feedback. In response to your suggestion, we have annotated the bond angles, distances, and key binding residues in Figure 4 and discussed them in the results section. Additionally, our study includes eight breast cancer-associated targets, not five. For the binding site descriptions, we have referenced the databases from which the targets were retrieved, as these sources provide comprehensive details on their topology and binding sites. We believe repeating this information in our manuscript would be redundant. However, we have included the resolution and PDB IDs of the targets around lines 118-121. In our methodology, we explained that the binding sites were selected based on known inhibitors, with their coordinates provided. Our analysis focuses on the residues in these regions that interact with our ligands (compounds), identifying key interacting residues within that known binding site to guide the further development of the lead compounds identified in our study.

Comment

The authors should provide MM_GBSA or PBSA energy calculations. It is advised to provide the dissected energy terms of these total energies (ΔGbind Lipo, ΔGbind Solv GB, ΔGbind vDW, ΔGbind Coulomb, and ΔGbind Ligand strain) to further evaluate the nature of interaction (dominant energy potential) based on these different energy terms. This would guide further hit to lead and lead optimization steps.

Response

Thank you for your suggestion and this important observation. Based on suggestions by other reviewers, we proceeded to do post-docking energy minimization to evaluate the stability of the protein-compound complexes. For individual residues contributing to energy, we have annotated residues that the ligands (compounds) interacted with and the type of bonds they formed. MM_GBSA or PBSA energy calculations would be valuable after MD simulation in our opinion but as said earlier, based on recommendations, we went ahead to do post-docking energy minimization which we have described at appropriate sections in the revised manuscript.

Comment

Based on the study results, what are the take-away messages. Authors are advised to highlight the future suggested structural modifications that would improve the hits’ predicted activities based on the computational findings. These insights would be beneficial for guiding future lead optimization and development.

Response

Thank you for your suggestion. We have made suggestions for the kind of approach that can be used for the structural optimization around lines 396-399. The key take-away messages have also been highlighted in the conclusion of the manuscript.

---

## [Decision Letter · Decision Letter 1]

9 Nov 2025

PONE-D-25-34728R1In silico evaluation of garlic-derived organosulfur compounds as multi-target inhibitors of breast cancer biomarkersPLOS ONE

Dear Dr. Danquah,

Thank you for submitting your manuscript to PLOS ONE. After careful consideration, we feel that it has merit but does not fully meet PLOS ONE’s publication criteria as it currently stands. Therefore, we invite you to submit a revised version of the manuscript that addresses the points raised during the review process.

Dear Authors,

Please revise your manuscript thoroughly in accordance with the reviewers’ comments, as incorporating these revisions is essential prior to publication.

If applicable, we recommend that you deposit your laboratory protocols in protocols.io to enhance the reproducibility of your results. Protocols.io assigns your protocol its own identifier (DOI) so that it can be cited independently in the future. For instructions see: https://journals.plos.org/plosone/s/submission-guidelines#loc-laboratory-protocols. Additionally, PLOS ONE offers an option for publishing peer-reviewed Lab Protocol articles, which describe protocols hosted on protocols.io. Read more information on sharing protocols at . Additionally, PLOS ONE offers an option for publishing peer-reviewed Lab Protocol articles, which describe protocols hosted on protocols.io. Read more information on sharing protocols at https://plos.org/protocols?utm_medium=editorial-email&utm_source=authorletters&utm_campaign=protocols..

We look forward to receiving your revised manuscript.

Kind regards,

Muhammad Umer Khan, Ph. D

Academic Editor

PLOS ONE

Journal Requirements:

Reviewers' comments:

Reviewer's Responses to Questions

**Comments to the Author**

1. If the authors have adequately addressed your comments raised in a previous round of review and you feel that this manuscript is now acceptable for publication, you may indicate that here to bypass the “Comments to the Author” section, enter your conflict of interest statement in the “Confidential to Editor” section, and submit your "Accept" recommendation.

Reviewer #2: (No Response)

Reviewer #3: (No Response)

2. Is the manuscript technically sound, and do the data support the conclusions?

Reviewer #2: (No Response)

Reviewer #3: Yes

3. Has the statistical analysis been performed appropriately and rigorously? 

Reviewer #2: (No Response)

Reviewer #3: Yes

4. Have the authors made all data underlying the findings in their manuscript fully available?

Reviewer #2: (No Response)

Reviewer #3: Yes

5. Is the manuscript presented in an intelligible fashion and written in standard English?

Reviewer #2: (No Response)

Reviewer #3: Yes

6. Review Comments to the Author

Reviewer #2: (No Response)

Reviewer #3: 1. In the Introduction section, the authors should include a brief overview of recent in silico techniques, supported by relevant references, to align with the title and provide better context for the study. This will enhance the reader’s understanding of the computational approaches used.

2. Docking requires 3D coordinates. Please state explicitly how the 2D PubChem records were converted to 3D (e.g., OpenBabel, RDKit, Avogadro, Chem3D, or a PubChem 3D conformer download) and include the software name and version.

3. Describe how protonation states (pH considered), stereochemistry, and tautomeric forms were handled before optimization and docking. Indicate whether explicit hydrogens were added and which protonation state was used.

4. Although MMFF94 and steepest descent are mentioned, the manuscript should specify the program or server used to perform the optimization (including version), the implementation of MMFF94, the maximum number of optimization steps or exact stopping criterion (rather than “varied until energy minimization was achieved”), and the final energy or RMS gradient reached.

5. Indicate the file formats generated (e.g., SDF, MOL2, PDB) and detail any further preparation steps used for docking (charge assignment, atom typing, conversion to PDBQT, conformer selection, number of conformers retained).

6. For reproducibility, please provide the optimized 3D structures (e.g., as an SDF/MOL2/PDB supplementary file) or a link to a repository containing these geometries.

7. The grid box parameters (center coordinates, box size, and grid spacing) are provided only for the Bcl-2 protein. Since docking was performed against eight target proteins, similar details should be included for all targets to ensure methodological completeness and reproducibility.

8. The manuscript describes post-docking energy minimization using OpenBabel and the Universal Force Field (UFF), which is appropriate for local structural relaxation. However, this step cannot substitute for molecular dynamics (MD) simulation, as energy minimization provides only a static refinement and does not account for time-dependent stability or conformational flexibility of the complex. It is recommended to perform MD simulations to more accurately assess the dynamic stability and binding behavior of the top compound–protein complexes.

9. Please include the accessible web links for all ADMET and drug-likeness tools for reproducibility and clarity.

10. The authors have discussed only the top-scoring complex in the docking evaluation. It is recommended to include docking results for all other ligand–protein complexes as supplementary data to provide a comprehensive view of binding affinities and interactions across all tested compounds.

7. PLOS authors have the option to publish the peer review history of their article (what does this mean?). If published, this will include your full peer review and any attached files.). If published, this will include your full peer review and any attached files.

.

Reviewer #2: **Yes:**Khaled DarwishKhaled Darwish

Reviewer #3: **Yes:**Hina ManzoorHina Manzoor

---

## [Author Response · Author response to Decision Letter 2]

4 Dec 2025

In each sample, the review comments are not italicized while the responses are italicized.

Reviewer 2

No responses received

Reviewer 3

Comment

1. In the Introduction section, the authors should include a brief overview of recent in silico techniques, supported by relevant references, to align with the title and provide better context for the study. This will enhance the reader’s understanding of the computational approaches used.

Response

Thank you for this important feedback. In the revised version of the manuscript, we have now provided a clearer rationale for the use of in silico approaches in cancer drug discovery and included a brief overview of key computational techniques, supported by relevant references. A notable example added is the use of in silico techniques to discover a novel inhibitor against HER2 and its subsequent experimental validation which proved its micropotency. This information has been incorporated into the Introduction to better align with the study’s focus and to help readers understand the significance and context of the computational methods employed. This can be found on lines 42-55.

Comment

2. Docking requires 3D coordinates. Please state explicitly how the 2D PubChem records were converted to 3D (e.g., OpenBabel, RDKit, Avogadro, Chem3D, or a PubChem 3D conformer download) and include the software name and version.

3. Describe how protonation states (pH considered), stereochemistry, and tautomeric forms were handled before optimization and docking. Indicate whether explicit hydrogens were added and which protonation state was used.

Response (to comments 2 and 3)

Thank you for your valuable feedback regarding our ligand preparation methodology. For the file formats and conversions needed for the molecular docking, we have updated the manuscript around lines 114 to 124 to clearly communicate it. We confirmed that all ligands were processed to ensure the correct major microspecies at physiological pH (7.4) using the OpenBabel pH flag (pH 7.4) during the SDF to MOL2 conversion. For stereochemistry, the (S) absolute configuration was chosen for cpd6, as this is the only chiral ligand and this stereoisomer is experimentally reported as potent, and for compound cpd1, the (Z) geometric isomer was maintained because it was experimentally shown to be potent in relevant studies. Finally, we employed ACD/ChemSketch to validate that the MOL2 files represented the most thermodynamically favorable tautomer. The structures incorporating these finalized definitions (protonation, stereochemistry, and tautomerism) are all depicted in Figure 1 of the revised manuscript around lines 121 to 123.

Comment

4. Although MMFF94 and steepest descent are mentioned, the manuscript should specify the program or server used to perform the optimization (including version), the implementation of MMFF94, the maximum number of optimization steps or exact stopping criterion (rather than “varied until energy minimization was achieved”), and the final energy or RMS gradient reached.

Response

Thank you for the comment. We now specify that ligand optimization was performed in OpenBabel (version 2.4.1) using MMFF94 with steepest descent for a maximum of 600 steps. We have updated our original statement of ‘ Geometry optimization was carried out using the MMFF94 force field and steepest descent (maximum 600 steps) algorithm with a convergence criterion of 1 × 10⁻⁷’ to ‘Geometry optimization was carried out using the MMFF94 force field and steepest descent (maximum 600 steps) algorithm with an RMS gradient convergence criterion of 1 × 10⁻⁷ after converting the SDF files to MOL2 formats considering the physiological pH of 7.4).

The RMS gradient convergence criterion of 1 × 10⁻⁷ is a very common and strict threshold used computation chemistry and physics for geometry optimizations. This update is now reflected in the Methods section (lines 122-123).

Comment

5. Indicate the file formats generated (e.g., SDF, MOL2, PDB) and detail any further preparation steps used for docking (charge assignment, atom typing, conversion to PDBQT, conformer selection, number of conformers retained).

Response

Thank you for your suggestion. The Structure preparation and file formats are now detailed in the updated manuscript around lines 114 to 160.

Comment

6. For reproducibility, please provide the optimized 3D structures (e.g., as an SDF/MOL2/PDB supplementary file) or a link to a repository containing these geometries.

Response

Thank you for the recommendation. The optimized structures can be accessed via https://doi.org/10.5281/zenodo.17804558. This has been included in the manuscript around lines 177-178.

Comment

7. The grid box parameters (center coordinates, box size, and grid spacing) are provided only for the Bcl-2 protein. Since docking was performed against eight target proteins, similar details should be included for all targets to ensure methodological completeness and reproducibility.

Response

Thank you for this important suggestion. For methodological completeness and reproducibility, we have updated the manuscript to include the grid box parameters of for all the eight target proteins and for cleaner and easier to read, we have put this in a table form as a supplemental data.

Comment

8. The manuscript describes post-docking energy minimization using OpenBabel and the Universal Force Field (UFF), which is appropriate for local structural relaxation. However, this step cannot substitute for molecular dynamics (MD) simulation, as energy minimization provides only a static refinement and does not account for time-dependent stability or conformational flexibility of the complex. It is recommended to perform MD simulations to more accurately assess the dynamic stability and binding behavior of the top compound–protein complexes.

Response

We greatly acknowledge your concern, but the decision to use post-docking energy minimization instead of full Molecular Dynamics (MD) simulation, as a reviewer in the initial round of review recommended, was based on computational efficiency and the immediate goal of structural refinement due to resource constraints. Minimization was used because it is significantly faster, making it the only feasible option for high-throughput analysis of the number of docked complexes we had. We used it to efficiently relieve high-energy steric clashes, and structural strains often present in docking poses, thereby ensuring the complex is geometrically valid and resides in the nearest local potential energy minimum. While MD offers complete dynamic and entropic analysis, minimization was a resource-conscious and sufficient method to validate the structural feasibility of the initial binding hypothesis. This limitation has been acknowledged in the revised manuscript around lines 457-462.

Comment

9. Please include the accessible web links for all ADMET and drug-likeness tools for reproducibility and clarity.

Response

We have cross-checked the links and updated them to ensure that they are accurate. This update is located around lines 161 – 170

Comment

10. The authors have discussed only the top-scoring complex in the docking evaluation. It is recommended to include docking results for all other ligand–protein complexes as supplementary data to provide a comprehensive view of binding affinities and interactions across all tested compounds.

Response

We appreciate your observation. However, the docking results for the other compounds have been provided in Table 1.

---

## [Decision Letter · Decision Letter 2]

24 Feb 2026

PONE-D-25-34728R2In silico evaluation of garlic-derived organosulfur compounds as multi-target inhibitors of breast cancer biomarkersPLOS One

Dear Dr. Danquah,

Thank you for submitting your manuscript to PLOS ONE. After careful consideration, we feel that it has merit but does not fully meet PLOS ONE’s publication criteria as it currently stands. Therefore, we invite you to submit a revised version of the manuscript that addresses the points raised during the review process.

If applicable, we recommend that you deposit your laboratory protocols in protocols.io to enhance the reproducibility of your results. Protocols.io assigns your protocol its own identifier (DOI) so that it can be cited independently in the future. For instructions see: https://journals.plos.org/plosone/s/submission-guidelines#loc-laboratory-protocols. Additionally, PLOS ONE offers an option for publishing peer-reviewed Lab Protocol articles, which describe protocols hosted on protocols.io. Read more information on sharing protocols at . Additionally, PLOS ONE offers an option for publishing peer-reviewed Lab Protocol articles, which describe protocols hosted on protocols.io. Read more information on sharing protocols at https://plos.org/protocols?utm_medium=editorial-email&utm_source=authorletters&utm_campaign=protocols..

We look forward to receiving your revised manuscript.

Kind regards,

Muhammad Umer Khan, Ph. D

Academic Editor

PLOS One

Journal Requirements:

Reviewers' comments:

Reviewer's Responses to Questions

**Comments to the Author**

1. If the authors have adequately addressed your comments raised in a previous round of review and you feel that this manuscript is now acceptable for publication, you may indicate that here to bypass the “Comments to the Author” section, enter your conflict of interest statement in the “Confidential to Editor” section, and submit your "Accept" recommendation.

Reviewer #3: All comments have been addressed

Reviewer #4: (No Response)

2. Is the manuscript technically sound, and do the data support the conclusions?

Reviewer #3: Yes

Reviewer #4: Yes

3. Has the statistical analysis been performed appropriately and rigorously? 

Reviewer #3: N/A

Reviewer #4: N/A

4. Have the authors made all data underlying the findings in their manuscript fully available?

Reviewer #3: Yes

Reviewer #4: Yes

5. Is the manuscript presented in an intelligible fashion and written in standard English?

Reviewer #3: Yes

Reviewer #4: Yes

6. Review Comments to the Author

Reviewer #3: The manuscript is acceptable in its current form and is recommended for publication. The study presents a well-structured and comprehensive in silico investigation, and the methodology applied is appropriate for the stated objectives. The results are clearly presented and supported by relevant analyses, providing meaningful insights into the interaction of the selected compounds with the proposed targets. Although molecular dynamics simulations were not performed, this does not detract from the overall scientific value of the work. Minor language and grammatical issues are present; however, they do not affect the clarity or interpretation of the findings. Overall, the manuscript meets the required standards for publication.

Reviewer #4: 1. The Introduction starts abruptly and lacks sufficient background context. The authors should begin with a brief overview of breast cancer burden, current therapeutic strategies, and existing clinical limitations to provide a logical foundation for the study.

2. The statement describing the identification of Z-ajoene and S-allyl-L-cysteine as promising candidates appears to report the study’s findings within the Introduction. Instead, the Introduction should focus on the rationale, knowledge gaps, and study objectives.

3. The manuscript lacks a clear justification for the selection of the proposed molecular targets in breast cancer. The authors should explicitly explain the biological and clinical relevance of these target proteins, including their roles in breast cancer progression, signaling pathways, and therapeutic relevance. A concise rationale for each target is required.

4. The authors should confirm whether the selected PDB structures correspond to human proteins. If any structures are derived from non-human homologs, the rationale for their selection and the level of structural or sequence similarity to human proteins should be clearly stated.

5. Appropriate references for each PDB ID should be provided. The authors are encouraged to cite the original structural studies (X-ray crystallography or cryo-EM publications) rather than only referring to the Protein Data Bank.

6. A comparative analysis with standard FDA-approved breast cancer drugs (e.g., tamoxifen, paclitaxel, doxorubicin, trastuzumab) should be included. Benchmarking docking scores, binding energies, and interaction profiles against standard therapeutics would improve the clinical relevance of the study.

7. There is inconsistent use of abbreviations throughout the manuscript. All abbreviations should be defined at first mention and used consistently. Unnecessary or redundant abbreviations should be avoided.

8. Potential off-target interactions and unintended biological effects of the proposed compounds should be discussed. Incorporating computational off-target prediction would strengthen the safety evaluation.

9. Molecular dynamics simulations should be performed for at least 200 ns to validate the docking results to ensure stability and convergence of the protein–ligand complexes.

10. A comparative discussion with previously published studies is missing. The authors should compare their findings with existing computational and experimental reports on similar targets and compounds, highlighting similarities, differences, and the novelty of the present work. This will help position the study within the current literature and clarify its scientific contribution.

7. PLOS authors have the option to publish the peer review history of their article (what does this mean?). If published, this will include your full peer review and any attached files.). If published, this will include your full peer review and any attached files.

.

Reviewer #3: **Yes:**Hina ManzoorHina Manzoor

Reviewer #4: **Yes:**Iqra KhurramIqra Khurram

---

## [Author Response · Author response to Decision Letter 3]

9 Mar 2026

Reviewer 4 Response

Comment 1

The Introduction starts abruptly and lacks sufficient background context. The authors should begin with a brief overview of breast cancer burden, current therapeutic strategies, and existing clinical limitations to provide a logical foundation for the study.

Response

Thank you for your comment. In response, we have revised the Introduction to provide clearer background context. Specifically, we now include a brief overview of the global burden of breast cancer (Lines 57-62), followed by discussion of current therapeutic strategies and their limitations, particularly the challenge of multidrug resistance.

Comment 2

The statement describing the identification of Z-ajoene and S-allyl-L-cysteine as promising candidates appears to report the study’s findings within the Introduction. Instead, the Introduction should focus on the rationale, knowledge gaps, and study objectives.

Response

Thank you for this comment. We agree that the Introduction should focus on the study rationale and knowledge gaps rather than report findings. Accordingly, the Introduction has been revised to remove statements suggesting study outcomes and instead emphasize the background, research gap, and study objectives.

Comment 3

The manuscript lacks a clear justification for the selection of the proposed molecular targets in breast cancer. The authors should explicitly explain the biological and clinical relevance of these target proteins, including their roles in breast cancer progression, signaling pathways, and therapeutic relevance. A concise rationale for each target is required.

Response

Thank you for this comment. The biological and clinical relevance of the selected molecular targets has been described in the Introduction (Lines 70-79). Specifically, we discuss their roles in key processes associated with breast cancer progression, including apoptosis regulation (Bcl-2 and XIAP), cell cycle control (CDK2 and CDK6), DNA replication and repair (topoisomerases I and II), and tumor angiogenesis (VEGFR2 signaling). In addition, the therapeutic relevance of G-quadruplex DNA structures in oncogene regulation is also highlighted.

Comment 4

The authors should confirm whether the selected PDB structures correspond to human proteins. If any structures are derived from non-human homologs, the rationale for their selection and the level of structural or sequence similarity to human proteins should be clearly stated.

Response

Thank you for this comment. We confirm that all selected PDB structures correspond to human proteins and are not derived from non-human homologs. To improve clarity, we have revised the Methods section to state this in the revised manuscript (Lines 140-141).

Comment 5

Appropriate references for each PDB ID should be provided. The authors are encouraged to cite the original structural studies (X-ray crystallography or cryo-EM publications) rather than only referring to the Protein Data Bank.

Response

We appreciate this suggestion. In the revised manuscript, around lines 141 to 145, we have referenced the publications which resolved the structure of the molecular targets in addition to their PDB ID.

Comment 6

A comparative analysis with standard FDA-approved breast cancer drugs (e.g., tamoxifen, paclitaxel, doxorubicin, trastuzumab) should be included. Benchmarking docking scores, binding energies, and interaction profiles against standard therapeutics would improve the clinical relevance of the study.

Response

Thank you for this suggestion. Comparative discussion with relevant FDA-approved breast cancer therapeutics has been included in the revised manuscript (Lines 398-425). Specifically, we compare the binding interactions and potential mechanisms of the identified compounds with established drugs such as palbociclib, Abemaciclib, etoposide, cediranib, and the G-quadruplex stabilizer quarfloxin. These comparisons highlight similarities in binding modes, interaction with key residues, and potential functional implications, while also acknowledging the relatively lower binding affinities of the garlic-derived compounds and the need for further structural optimization.

Comment 7

There is inconsistent use of abbreviations throughout the manuscript. All abbreviations should be defined at first mention and used consistently. Unnecessary or redundant abbreviations should be avoided.

Response

Thank you for this comment. We have carefully reviewed the manuscript to ensure that all abbreviations are defined at their first occurrence and used consistently throughout the text. Any redundant or unnecessary abbreviations were also checked and corrected where appropriate to improve clarity.

Comment 8

Potential off-target interactions and unintended biological effects of the proposed compounds should be discussed. Incorporating computational off-target prediction would strengthen the safety evaluation.

Response

Thank you for this suggestion. In response, we evaluated the potential off-target interactions and unintended biological effects of the proposed compounds through computational ADMET and toxicity predictions. The analysis indicated that P-glycoprotein may be inhibited by compounds cpd2, cpd3, cpd4, and cpd5, while CYP2C9 inhibition was predicted for cpd1 (Z-ajoene). These findings and their potential implications have been discussed in the revised manuscript (Lines 450-458).

Comment 9

Molecular dynamics simulations should be performed for at least 200 ns to validate the docking results to ensure stability and convergence of the protein–ligand complexes.

Response

Thank you for this suggestion. We agree that molecular dynamics (MD) simulations can provide important information into the stability and convergence of protein-ligand complexes. However, due to logistical and computational constraints, we were unable to perform extended MD simulations during this study. As an alternative, post-docking energy minimization was conducted to refine the docking poses and improve the stability of the predicted complexes. We acknowledge the absence of MD simulations as a limitation of the present study and have explicitly stated this in the manuscript.

Comment 10

A comparative discussion with previously published studies is missing. The authors should compare their findings with existing computational and experimental reports on similar targets and compounds, highlighting similarities, differences, and the novelty of the present work. This will help position the study within the current literature and clarify its scientific contribution.

Response

Thank you for this great suggestion. We agree that positioning our findings within the context of existing literature strengthens the scientific contribution of the study. Accordingly, we have revised the Discussion section (Lines 426-441) to include comparative analysis with previously published computational and experimental studies involving similar molecular targets and natural compounds. These additions highlight similarities and differences in binding interactions, docking affinities, and reported biological activities, while also emphasizing the novelty and potential therapeutic relevance of the garlic-derived compounds investigated in this study.

---

## [Decision Letter · Decision Letter 3]

15 Apr 2026

In silico evaluation of garlic-derived organosulfur compounds as multi-target inhibitors of breast cancer biomarkers

PONE-D-25-34728R3

Dear Dr. Danquah,

We’re pleased to inform you that your manuscript has been judged scientifically suitable for publication and will be formally accepted for publication once it meets all outstanding technical requirements.

An invoice will be generated when your article is formally accepted. Please note, if your institution has a publishing partnership with PLOS and your article meets the relevant criteria, all or part of your publication costs will be covered. Please make sure your user information is up-to-date by logging into Editorial Manager at Editorial Manager® and clicking the ‘Update My Information' link at the top of the page. For questions related to billing, please contact  and clicking the ‘Update My Information' link at the top of the page. For questions related to billing, please contact billing support..

Kind regards,

Muhammad Umer Khan, Ph. D

Academic Editor

PLOS One

Additional Editor Comments (optional):

Reviewers' comments:

Reviewer's Responses to Questions

**Comments to the Author**

1. If the authors have adequately addressed your comments raised in a previous round of review and you feel that this manuscript is now acceptable for publication, you may indicate that here to bypass the “Comments to the Author” section, enter your conflict of interest statement in the “Confidential to Editor” section, and submit your "Accept" recommendation.

Reviewer #4: All comments have been addressed

2. Is the manuscript technically sound, and do the data support the conclusions?

Reviewer #4: Yes

3. Has the statistical analysis been performed appropriately and rigorously? 

Reviewer #4: N/A

4. Have the authors made all data underlying the findings in their manuscript fully available?

Reviewer #4: Yes

5. Is the manuscript presented in an intelligible fashion and written in standard English?

Reviewer #4: Yes

6. Review Comments to the Author

Reviewer #4: All comments have been thoroughly and carefully addressed, and the manuscript has been revised accordingly. It is now accepted in its current form.

7. PLOS authors have the option to publish the peer review history of their article (what does this mean?). If published, this will include your full peer review and any attached files.). If published, this will include your full peer review and any attached files.

.

Reviewer #4: **Yes:**Iqra KhurramIqra Khurram

---

## [Editor Report · Acceptance letter]

PONE-D-25-34728R3

PLOS One

Dear Dr. Danquah,

I'm pleased to inform you that your manuscript has been deemed suitable for publication in PLOS One. Congratulations! Your manuscript is now being handed over to our production team.

Kind regards,

on behalf of

Dr. Muhammad Umer Khan

Academic Editor

PLOS One